# Incremental Validity of Trait Impulsivity, Dysfunctional Emotional Regulation, and Affect Lability in the Predictions of Attention Deficit Hyperactivity Disorder and Oppositional Defiant Disorder Symptoms in Adults

**DOI:** 10.3390/bs14070598

**Published:** 2024-07-14

**Authors:** Rapson Gomez, Stephen J Houghton

**Affiliations:** 1School of Health and Biomedical Sciences, Federation University, Melbourne, VIC 3000, Australia; rapson.gomez@federation.edu.au; 2Graduate School of Education, University of Western Australia, Perth, WA 6009, Australia

**Keywords:** trait impulsivity, emotional regulation, affect lability, ADHD, ODD, incremental validity

## Abstract

Difficulties in emotion regulation (DER) is a defining feature of attention deficit hyperactivity disorder (ADHD), and arguments are being made for it to be considered as a defining feature of oppositional defiant disorder (ODD). However, the consensus is that it is better viewed as an important correlate distinct from ADHD. This study examined the incremental validity of DER over and above trait impulsivity (TI) in the predictions of ADHD and ODD symptoms. It also examined the incremental validity of affect lability (AL) over and above TI and (DER) in these predictions. Five hundred and twenty-five adults from the general community completed a series of questionnaires. A model-based SEM approach for evaluating incremental validity indicated that TI predicted ADHD and ODD symptoms over age; DER predicted ADHD and ODD symptoms over age and TI; and AL did not predict ADHD and ODD symptoms over and above age, IT, or DER. In addition, AL predicted ADHD and ODD symptoms over age and TI, and DER also predicted ADHD and ODD symptoms over and above age, TI, and AL. In conclusion, TI is core to ADHD, and although DER is important, it is unlikely to be relevant as a diagnostic indicator for ADHD or ODD. These findings notwithstanding, there is need for caution when interpreting our findings, as the study did not control for potentially influencing factors on emotional regulation such as age, gender, culture, and existing psychopathologies.

## 1. Introduction

According to the latest edition of the Diagnostic and Statistical Manual of Mental Disorders [1], the core symptoms of attention deficit hyperactivity disorder (ADHD) are inattention (IA; nine symptoms) and hyperactivity/impulsivity (HI; nine symptoms). For oppositional defiant disorder (ODD), there are eight symptoms. ADHD and ODD are closely related. In the trait impulsivity hypothesis (TIH) [2], both these disorders have been linked to trait impulsivity (TI). Some experts have proposed that difficulties in emotion regulation are another defining feature of ADHD [3,4,5,6], especially in adults [7,8,9]. Although not as robust, arguments have been made for difficulties in emotion regulation to be considered as a defining feature of ODD [10,11,12]. However, at present, the consensus is that difficulties in emotion regulation are better viewed as an important correlate that is distinct from ADHD [13] and ODD [11,12]. Given this disparity in views, the major goal of the current study was to examine whether support exists for the incremental validity of difficulties in emotion regulation over and above TI in the prediction of ADHD symptoms (IA and HI) and ODD symptoms.

According to TIH [2,14], a highly heritable, latent individual difference TI factor confers vulnerability to externalizing disorders, such as ADHD and ODD. At extreme levels, this is expressed as a preference for immediate rewards over larger delayed rewards, actions taken without forethought, failures to plan ahead, and deficiencies in self-control [2].

Developmentally, this liability contributes first to the development of HI symptoms (during preschool years); followed soon after by ODD symptoms (during early childhood); and, subsequently, conduct disorder (during the middle school period), substance use disorder (during adolescence), and, lastly, antisocial personality disorder (in young adulthood) [2,15]. The TIH postulates that ADHD inattention (IA) symptoms develop by school entry and are secondary to HI symptoms [16]. Furthermore, the development of HI and ODD depends on exposure to neurocognitive and environmental risk factors [2], the key ones being difficulties in emotional regulation and exposure to coercive family processes and/or deviant peer group affiliation [2,17]. Overall, therefore, although TI, HI, ODD, IA, neurocognitive, and environmental risk factors are intertwined, the relationship is complex, following a developmental sequence beginning from TI and then extending to HI, followed by ODD symptoms. Related to this sequence, the emergence of IA is dependent on the presence or absence of neurocognitive and environmental risk factors. In this respect, difficulties in emotional regulation and exposure to coercive family processes and/or deviant peer group affiliation are critical risk factors.

Difficulties in emotion regulation refers to difficulties in attending to, understanding, modifying, and responding to one’s generally negative emotional state in order to ensure an adaptive goal-oriented response that is functional and adaptive [18]. The ADHD and ODD literature depicts a separation of difficulties in emotion regulation into emotional or affect lability (AL) and dysfunctional emotion regulation (DER). AL refers to the threshold, intensity, and duration of affective arousal to (mostly) negative emotions, whereas DER refers to problems associated with the processes involved in dealing with emotional reactivity in an effective and flexible manner in order to facilitate adaptive functioning [4,19]. We adopt this separation in this paper. However, to avoid confusion, in this paper, the term “difficulties in emotional regulation” refers to both AL and DER, and DER is used to refer to dysfunctional emotion regulation (i.e., problems associated in the processes involved in dealing with AL in an effective and flexible manner to facilitate adaptive functioning).

In a qualitative review [13], it was concluded that 34–70% of adults with ADHD also experience emotional dysregulation problems. Existing studies show that both DER [13,20] and AL [8,21,22,23] are positively and strongly associated with ADHD. Based on the proposed model [24], a meta-analysis [19] involving 77 studies on children and adolescents (*N* = 32,044) concluded that ADHD was associated with the greatest impairment with AL (weighted effect size *d* = 0.95), followed closely by DER (weighted effect size *d* = 0.80), and less so with empathy/callous–unemotional traits (weighted effect size *d* = 0.68) and emotion recognition/understanding (weighted effect size *d* = 0.64). Another meta-analysis [4] that involved 13 studies of adults (*N* = 2535) found that AL and DER have strong associations with ADHD (weighted effect size Hedges’g = 1.30, and Hedges’g = 1.17, respectively). Overall, therefore, existing data indicate that ADHD is most strongly associated with AL and DER, with the association being slightly stronger for AL.

Compared to ADHD, fewer studies have examined how difficulties in emotion regulation are associated with ODD (e.g., [10,11,12,25,26,27]). These studies, all involving children, have shown strong and positive associations between difficulties in emotion regulation and ODD. One study [27] found this for both AL and DER with ODD. Relatedly, Ref. [10] suggested that ODD is better considered as a disorder of emotional regulation. To date, however, there appears to be no study examining the association between emotional regulation problems and ODD in adults. Nevertheless, considering that the available literature indicates close comparability between childhood ODD and adult ODD in virtually all respects, it is conceivable that adult ODD would also be associated with difficulties in emotional regulation.

Despite existing findings showing associations of DER and AL with ADHD and ODD, we believe that these findings are limited. First, given that TI, AL, and DER are closely related [28], it is conceivable there will be many shared variances between them. However, past studies have not considered this, and therefore, it is uncertain what the findings from past studies show in terms of the unique relations of the variables involving difficulties in emotional regulation with ADHD and ODD. Testing for incremental validity may provide a way to bring clarity to this. For instance, examining whether difficulties in emotional regulation provide additional variance to the predictions of IA, HI, and ODD, after accounting for TI, will inform us regarding the relevance of difficulties in emotional regulation to the predictions of IA, HI, and ODD after accounting for TI. Similarly, by examining whether AL offers incremental variance over DER or, conversely, if DER offers incremental variance over AL after accounting for TI will inform us about the relative importance of AL versus DER in predicting IA, HI, and ODD. Such knowledge can help to clarify intervention targets.

Another limitation is that past studies have focused on ADHD overall, and little attention has been given to the IA and HI dimensions. Consequently, there is little understanding regarding how difficulties in emotional regulation are associated with IA and HI separately. As the DSM-5 suggests, there are three presentations of ADHD, namely, inattentive type (having mostly IA symptoms), hyperactive/impulsivity type (having mostly HI symptoms), and combined type (having both IA and HI symptoms). Such knowledge can be expected to provide more specific guidance for the targeted assessment and treatment of the different ADHD presentations.

A third limitation is that virtually all previous studies in this area pertaining to ODD have involved children. This is not surprising because ODD is considered a childhood disorder in the DSM-5. However, recent findings which applied the same ODD symptoms to adults found that they persist into adulthood and that they are associated with functional maladjustments, including greater social impairment, friendship problems, online antagonistic behavior, and conflict with (non-parental) authority figures [29]. As in children, ODD is highly comorbid with ADHD in adults [30,31]. A community survey [32] reported the lifetime prevalence of ODD to be approximately 10% (a risk consistent across five cohorts between the ages of 18 and 44 years), with ODD showing high rates of comorbidity with ADHD (35%), substance misuse disorders (47%), and impulse control disorders (68%). Thus, it can be argued that, like ADHD, ODD is relevant to adults. Relatedly, understanding the incremental variance of AL and DER after accounting for TI in the prediction of ODD in adults is also an important empirical question. As in ADHD, such knowledge can be expected to provide more specific guidance for targeted assessment and treatment of ODD in adults.

Overall, therefore, a carefully designed study aimed at examining the incremental validity of DER over and beyond TI, as well as AL over and beyond TI and DER, in the prediction of IA, HI, and ODD is important. Moreover, comparing the appropriate incremental validity values of this model with another incremental validity analysis of AL over and beyond TI, and DER over and beyond TI and AL, after controlling TI in the prediction of IA, HI, and ODD will provide a comprehensive understanding of how emotional regulation difficulties (within and across DER and AL) are related to IA, HI, and ODD. For instance, it will provide insights related to the question of whether DER and/or AF are core or diagnostic indicators of ADHD and ODD, and which components of difficulties in emotional regulation are more important in explaining IA, HI, and ODD.

Traditionally, incremental validity is generally assessed using hierarchical regression analyses. In this method, based on the study’s aims, the predictors are entered in a prespecified order in different steps. The *R*^2^ change for one step to the next step is computed, and the corresponding *F*-statistic is tested for significance. A significant *F* is indicative that the predictor (or block of predictors) in that step has incremental validity above and beyond the predictor (or block of predictors) entered in the previous step(s). Although this method is generally straightforward and easy to implement and interpret, it has its limitations. According to [33], it is a multistep method that is cumbersome. Additionally, as the predictors and outcome are observed variables, they include random errors and the independence of the constructs cannot be assured, thereby leading to possibilities of bias in the findings. Also, it is not flexible, meaning it cannot examine multiple outcomes simultaneously. To overcome the limitations of the regression approach, [33] developed and demonstrated a new structural equation modeling (SEM) method to investigate incremental validity, the details of which are provided in Section 2.

Given the existing limitations in this area of research, the major aim of the present study was to examine the issues raised earlier through two separate incremental validity analyses. In the first analysis (incremental validity analysis 1), we examined whether DER offered incremental variance over TI in the prediction of IA, HI, and ODD, and whether AL offered incremental variance over TI and DER in the prediction of IA, HI, and ODD. In the second analysis (incremental validity analysis 2), we examined whether AL offered incremental variance over TI in the prediction of IA, HI, and ODD, and whether DER offered incremental variance over TI and AL in the prediction of IA, HI, and ODD. Both analyses (1 and 2) were conducted used the SEM approach proposed by [33] for evaluation of incremental validity on the same group of adults from the general community. In general, we expected TI to be associated with IA, HI, and ODD and for DER and/or AF to show incremental validity in the predictions of IA, HI, and ODD. However, in the absence of data, we made no specific predictions about the incremental validity of AL and DER in the predictions of IA, HI, and ODD after controlling for TI, nor for AL over TI and DER or for DER over TI and AL. Despite this, we expected TI to be associated with IA, HI, and ODD and for DER and/or AF to show incremental validity in the predictions of IA, HI, and ODD above and beyond TI.

## 2. Materials and Methods

### 2.1. Participants

In total, 525 adults (142 males, mean age = 34.74 years, *SD* = 12.91 years and 383 females, mean age = 32.60 years, *SD* = 12.95 years) from the Australian general community, who were aged 18 to 65 years (mean age = 32.91 years, *SD* = 12.93 years), were recruited to participate in the study. There was no significant difference between males and females in age, *t* (530) = 0.898, *p* = 0.370. The majority of the participants were either employed full-time or students, had completed higher education (university education), and were in some sort of relationship. Appendix A provides background information on the participants. As shown in this table, based on symptom counts of the recoding ADHD items (in the CSS) in terms of ratings of 0 and 1 as symptoms absent and 2 and 3 as symptoms present, as well as the threshold number (5 for more symptoms in the respective ADHD symptom groups), 61 individuals (11.62%) met the ADHD symptom requirement. Of these, 25 (4.76), 11 (2.1), and 25 (4.76) had predominantly inattentive-type ADHD, predominantly hyperactive/impulsive-type ADHD, and combined-type ADHD.

Soper’s software [34] for computing sample size requirements for CFA models was used to evaluate the sample size requirement for the present study. For this, the anticipated effect size was set at 0.3, the power at 0.8, the number of latent variables at 5 (i.e., TI, AL, DER, IA and HI), the number of observed variables at 26 (5 indicators for TI, 5 indicators for DER, 3 indicators for AL, 9 indicators for IA, and 9 indicators for HI), and probability at 0.05. The analysis recommended a minimum sample size of 233. As our sample size (*N* = 525) exceeded this recommendation, sufficient power for the study can be assumed.

### 2.2. Measures

All participants completed a demographic information questionnaire that asked for age, gender, education, employment and relationship status, and previous diagnosis of ADHD and ODD. They also completed the following self-rating questionnaires.

### 2.3. Current Symptom Scale [35]

ADHD symptom ratings were obtained using the CSS. The CSS includes the 18 DSM-IV/DSM-IV-TR ADHD symptoms and the 8 ODD DSM-IV/DSM-IV-TR symptoms. These correspond to DSM-IV IA, HI, and ODD symptoms. These symptoms are comparable to the DSM-5 symptoms for ADHD. For each symptom, participants indicate the frequency of symptoms over the previous six months on a four-point scale ranging from 0 (“never or rarely”) to 3 (“very often”). Higher scores represent greater severity. The Cronbach’s alpha values for the IA, HI, and ODD symptom groups in the current study were 0.89, 0.83, and 0.85, respectively. (The CSS is copyrighted, and therefore not reproduced in the Appendix A).

### 2.4. Short-Urgency-Premeditation-Perseverance-Sensation Seeking- Positive Urgency scale (S-UPPS-P) [36]

The 20-item Short Urgency–Premeditation–Perseverance–Sensation Seeking–Positive Urgency scale (S-UPPS-P0) [36] was used to measure TI. It has dimensions covering sensation seeking, lack of premeditation, lack of perseverance, negative urgency, and positive urgency. Taken together, these dimensions appear to conceptualize impulsivity as hypothesized for the temperament liability factor in the TIH (i.e., behaviorally rash responses that lack planning and premeditation, as well as difficulties in delayed gratification). Each item is rated using a four-point Likert scale ranging from 1 (“agree strongly”) to 4 (“disagree strongly”) in terms of how an act/incident described in it applies to them during the last 6 months. Higher scores indicate greater impulsivity. In the present study, the Cronbach’s alpha value for the full scale was 0.86. A copy of the S-UPPS-P is provided in the Appendix A.

### 2.5. The Affect Liability Scale-18 (ALS-18) [37]

The ALS, a self-report measure, was used to measure AL. Developed from its longer 58-item version [38], the ALS-18 measures AL in terms of changeability in mood between three affect states: anxiety/depression, depression/elation, and normal/anger. Each item is scored on four-point scale ranging from 0 (“very uncharacteristic of me”) to 3 (“very characteristic of me”) in terms of how often the items apply to the individual. Higher scores indicate more severe AL. In the present study, the Cronbach’s alpha value was 0.97 for the ALS-18. A copy of the ALS-18 is provided in the Appendix A.

### 2.6. Difficulties in Emotional Regulation Strategies-36

The 36-item Difficulties in Emotional Regulation Strategies scale-36 (DERS-36) [39] was used to measure DER. The DERS-36 is a multidimensional measure, with scales for non-acceptance (i.e., nonacceptance of emotional responses), goals (i.e., difficulties engaging in goal-directed behavior), impulse (i.e., impulse control difficulties), awareness (i.e., lack of emotional awareness), strategies (i.e., limited access to emotion regulation strategies), and clarity (i.e., lack of emotional clarity). The content of the items for the non-acceptance factor includes the experience of negative secondary emotional responses and non-accepting reactions to distress, such as shame, guilt, and self-blame for one’s negative emotions. For goals, the items include difficulties in concentration and accomplishing tasks when upset. Impulse items include difficulties controlling one’s behavior when distressed and feeling overwhelmed. Awareness includes items covering paying attention and acknowledging emotions. Strategies include items covering a belief that there is nothing that can help regulate the negative emotions being experienced (i.e., hopelessness). Clarity includes items covering the ability to understand emotions. As the awareness subscale has shown problems of reliability and validity [40,41], it has been suggested that this subscale be omitted. This suggestion was adopted here. Of importance, the content of the DERS-36 is not contaminated with AL items. For the DERS-36, higher scores indicate greater DER. In the present study, the Cronbach’s alpha value was 0.97 for the DERS-36. A copy of the DERs-36 is provided in the Appendix A.

### 2.7. Procedure

Approval for the study was obtained from the Human Research Ethics Committee of Federation University (Australia; ID no: 2543, Protocol no: 2014/058; internal reference no: A14-058). Following this, the study was advertised widely on the approved University’s noticeboards and on social media (Facebook), the Australian Psychology Society’s website, and on general community areas such as bus stops. Participants were recruited online over a 2-month time period using Survey Monkey. An electronic survey link was provided for participants to access and complete the questionnaires, which were counterbalanced. Proceeding with the survey was taken as informed consent to participate. Participants recruited via the Federation University psychology participant pool received research participation credit, whereas other participants did not receive any incentive for participation. Furthermore, the authors report there are no competing interests to declare.

Four participants had missing values for the entire S-UPPP-P, and another three had missing values for the entire ALS-18. These seven participants were excluded from the analyses. For simplicity, we used the plausible values on IA, HI, and ODD as individual measured outcomes (for details on plausible values see [42]), and all the item responses were treated as continuous variables. With the proper model constraints imposed, the Γ matrix includes the semi-partial correlations that can be used to facilitate inference regarding the incremental validity of the latent constructs.

### 2.8. Statistical Analysis

In the model-based SEM method proposed by [33] for evaluating incremental validity, the predictors can be measured variables or latent constructs, and they can be studied as individual predictors or in blocks predicting multiple variables. To this end, they also provide technical details underpinning four different scenarios, where the predictors of interest are: (a) individual measured predictors, (b) individual latent predictors, (c) blocks of measured predictors, and (d) blocks of latent predictors. Empirical examples (including the appropriate M*plus* and R syntax) illustrating the first and second scenarios were also presented. Of these, the second example is particularly relevant to the current study as it was used as a template to conduct our analysis.

Incremental validity is only theoretically meaningful when there is support for the discriminant validity of the measurement model [33]. Therefore, CFA was initially used to examine whether there was support for a three-factor model for discriminant validity. The latent factors for the three-factor model were TI, DER, and AL. Latent TI was modeled by the five subdomains in the UPPPS-P (sensation seeking, lack of premeditation, lack of perseverance, negative urgency, and positive urgency). Latent DER was modeled by five domains in the DERS-36 (goals, impulsivity, awareness, clarity, and strategies), and latent AL was modeled by the three domains in the ALS-18 (anxiety/depression, depression/elation, and normal/anger). This model is depicted as Appendix A.

Model fit was evaluated using the root mean squared error of approximation (RMSEA), comparative fit index (CFI), and Tucker–Lewis index (TLI). According to [43], RMSEA values < 0.06 = good fit, <0.08 = acceptable fit, and >0.08 to 0.10 = marginal fit. For CFI and TLI, values ≥ 0.95 = good fit and ≥0.90 = acceptable fit. For the SRMR, the value should be less than 0.05 for a good fit [43], although values smaller than 0.10 may be interpreted as acceptable [44]. A two-index approach was recommended [45] for evaluating model fit that includes fit in terms of the SRMR value and either the TLI, CFI, or RMSEA. For the current study, a model was considered acceptable if the SRMR value was smaller than 0.10 and if either TLI or CFI showed acceptable fit or the RMSEA has at least a marginal fit. For support of discriminant validities, the correlations of the latent factors in the model had to be ≤0.85 [46].

Contingent on support for the discriminant validities of the factors in the three-factor model, we used the M*plus* syntax supplied by [33] to assess the incremental validity of the three focal latent constructs (TI, DER, AF). The structural model for incremental validity analysis 1 is shown in Figure 1, with the order of entry as age, TI, DER, and AL. As will be noticed, the analyses involved one observed covariate (Age) and three latent predictors (trait impulsivity, dysfunctional emotional regulation, and affect lability), as well as three latent outcomes (IA, H, and ODD). Age was included as a covariate because it showed significant negative correlations with all the predictor and outcome variables (see results section and Appendix A). Also, past studies have generally shown that scores on the DERS and AL decrease with age [47,48,49]. As for gender, the current study (see Appendix A) and past studies show that DERS scores are unrelated to gender [39,50,51,52]. Therefore, gender was not included as a covariate. The model shown in Figure 1 assessed the incremental validity for trait impulsivity, dysfunctional emotional regulation, and affect lability in predicting IA, HI, and ODD after controlling for the corresponding preceding predictors. With the proper model constraints imposed, the path coefficients (Γ matrix) contained the semi-partial correlations that could be interpreted as the statistical evidence for incremental validity for trait impulsivity, dysfunctional emotional regulation, and affect lability. For incremental analysis 2, the order of entry was age, TI, AL, and DER. Age was entered first, as it is a background variable. TI was entered next, as it is a precursor for IA, HI, and ODD [2]. In the second incremental validity model, the dysfunctional emotional regulation was replaced by affect lability, and the affect lability was replaced by dysfunctional emotional regulation. All other variables and paths remained unchanged.

In brief, the model is specified as follows. The four individual latent predictors are arranged in the designated order. Each latent predictor is regressed on its preceding latent variables, and the residualized portion is modeled with the exogenous phantom variables e1 to e4, respectively, which are all constrained to have unit variance. The factor loadings for TI, DER, and AF are constrained to be non-negative. The latent predictors (TI, DER, and AF) are constrained to have unit variance, and its loadings are freely estimated. In this model, the path coefficients from e1 to e4 with the outcomes (IA, HI and ODD) represent the semi-partial correlations between the focal predictors (age, TI, DER, and AF) and the effects of the preceding variables removed from the predictors, respectively.

The elements in the Γ matrix (semi-partial correlations) in the output for the analysis can be used to examine the change in *R*^2^ value (∆*R*^2^). The point estimate is obtained by squaring the elements in the Γ matrix, and the confidence interval is built either based on asymptotic normality or using bootstrapping. For the current study, the confidence intervals for these point estimates were computed using the free software calculator for ∆*R*^2^, downloaded from https://www.danielsoper.com/statcalc/calculator/aspx?id=28 (accessed on 20 October 2022). Although there were multiple comparisons involved, we did not adjust for this, following the tutorial example in [2].

## 3. Results

### 3.1. Descriptives and Correlations of Study Variables

Appendix A shows the mean scores, *SDs*, and the intercorrelations of all the study variables. As shown, the IT, DER, AF, IA, HI, and ODD scores correlated significantly and positively with each other. All of these variables correlated significantly and negatively with age and showed no significant associations with gender.

### 3.2. Factor Structure

The initial fit values for the CFA that examined support for the three-factor model, with latent factors for TI, difficulties in emotional regulation, and affect lability, are shown in Table 1. The table also includes the fit values for a one-factor model in which all the indicators for TI, difficulties in emotional regulation, and affect lability were loaded on a single factor. As can be seen, the fit values for the one-factor model all indicted poor fit. For the three-factor model, the fit values were: χ^2^ = 369.675, *df* = 62, *p* < 0.001; CFI = 0.889; TLI = 0.861; RMSEA = 0.097 (90% confidence interval = 0.088/0.107); and SRMR = 0.062. Therefore, based on the cut-off scores used for evaluating model fit, the fir was at least acceptable for this model. In this model, the correlations of TI with difficulties in DER and AF were 0.689 and 0.662, respectively. The correlation between DER and AF was 0.815. As all of these values were less than 0.85, discriminant validity between these constructs can be assumed [46] (see also Appendix A).

### 3.3. Incremental Validity

#### 3.3.1. Incremental Validity Analysis 1

Table 2 shows the parameter estimates (and corresponding robust standard errors) for the Γ matrix in the SEM for incremental validity analysis 1. The rows contain path coefficient estimates for IA, HI, and ODD, with each column corresponding to each of the four predictors in the prespecified order. Based on the results in the third column, it can be inferred that TI was positively and strongly predictive of IA, HI, and ODD after controlling for age (*z* = 16.268, *z* = 12.744, and *z* = 14.692, respectively). In the fourth column, DER predicted IA, HI, and ODD beyond age and TI (*z* = 5.431, *z* = 6.059 and *z* = 7.191, respectively). In the last column, it is shown that AF was not associated with all three outcomes (IA, HI, and ODD) when controlling for age, TI, and DER (*z* = 0.816, *z* = 1.320, and *z* = 0.488, respectively), i.e., not adding additional variance explaining all three outcomes.

Using these semi-partial correlations, we computed the relevant ∆*R*^2^, and these are also presented in Table 3. As shown in the table, TI accounted for 44.5% of the variability in IA (∆*R*^2^ = (0.667)^2^ = 0.445, 95% bootstrap CI [0.382/0.508]); 30.0% of the variability in HI (∆*R*^2^ = (0.548)^2^ = 0.312, 95% bootstrap CI [0.247/0.377]); and 32.8% of the variability in ODD (∆*R*^2^ = (0.573)^2^ = 0.328, 95% bootstrap CI [0.263/0.393]) after controlling for age. DER had an additional, albeit relatively lower, contribution, explaining only 7.7% of the variability in IA (∆*R*^2^ = (0.277)^2^ = 0.077, 95% bootstrap CI [0.034/0.120]), 9.7% of the variability in HI (∆*R*^2^ = (0.309)^2^ = 0.097, 95% bootstrap CI [0.048, 0.14[2]), and 11.4% of the variability in ODD (∆*R*^2^ = (0.338)^2^ = 0.114, 95% bootstrap CI [0.063, 0.165]) after controlling for TI and age. AL had no additional significant contribution, accounting for only 0.1% of the variability in IA (∆*R*^2^ = (0.040)^2^ = 0.001, 95% bootstrap CI [−0.004, 0.006]), 0.4% of the variability in HI (∆*R*^2^ = (0.066)^2^ = 0.004, 95% bootstrap CI [−0.002, 0.015]), and 0.0% of the variability in ODD (∆*R*^2^ = (0.020)^2^ = 0.000, 95% bootstrap CI [−0.004, 0.002]) after controlling for DER, TI, and age.

#### 3.3.2. Incremental Validity Analysis 2

Table 3 shows the parameter estimates (and corresponding robust standard errors) for the Γ matrix for incremental validity analysis 2. As shown, the results for columns 2 and 3 are identical to those in incremental validity analysis 1. Thus, no comment is necessary on the findings in these columns. With reference to the findings shown in column 4 (incremental validity of AL above and beyond age, TI), AL predicted IA, HI, and ODD beyond age and TI (*z* = 4.840, *z* = 2.655, and *z* = 4.694, respectively). For column 5 (incremental validity of AL above and beyond age, TI, AL), DER predicted IA, HI, and ODD (*z* = 0.400, *z* = 5.833, and 5.905, respectively) above and beyond age, TI, and AF.

AL had an additional contribution, explaining only 4.3% of the variability in IA (∆*R*^2^ = (0.208)^2^ = 0.043, 95% bootstrap CI [0.009/0.077]), 2.1% of the variability in HI (∆*R*^2^ = (0.146)^2^ = 0.021, 95% bootstrap CI [−0.003, 0.045]), and 5.3% of the variability in ODD (∆*R*^2^ = (0.230)^2^ = 0.053, 95% bootstrap CI [0.016/ 0.090]) after controlling for TI and age. DER contributed 3.5% (∆*R*^2^ = (0.188)^2^ = 0.035, 95% bootstrap CI [0.004/0.066]), 7.8% (∆*R*^2^ = (0.280)^2^ = 0.079, 95% bootstrap CI [0.034/0.122]), and 6.2% (∆*R*^2^ = (0.248)^2^ = 0.062, 95% bootstrap CI [0.022/0.102]) to the predictions of IA, HI, and ODD, respectively, after controlling for AF, TI, and age.

## 4. Discussion

The major aim of the present study was to examine the incremental validity of DER and AL over and above TI in the prediction of IA, HI, and ODD. Related to this aim, the study examined the incremental validity of AL over and above TI and DER in the prediction of IA, HI, and ODD (incremental validity analysis 1) and the incremental validity of DER over and above TI and AL in the prediction of IA, HI, and ODD. Initially, we examined support for a three-factor structural model with oblique factors for TI, DER, and AF, and our initial analysis indicated support for this, including the discriminant validity of the three factors in the model. In brief, our findings in incremental validity analysis 1 showed that TI predicted IA, HI, and ODD over age; DER predicted IA, HI, and ODD over age and TI; and AL did not predict IA, HI, or ODD over and above age, IT, and DER. Our findings for incremental validity analysis 2 showed that TI predicted IA, HI, and ODD over age; AL predicted IA, HI, and ODD over age and TI; and DER also predicted IA, HI, and ODD over and above age, IT, and AL.

Considering TI and difficulties in emotional regulation, the amount of variance predicated by TI was medium (44.5%, 30.0%, and 32.8% of the variability for IA, HI, and ODD, respectively), whereas the amount of variance predicated by DER was low (7.7%, 9.7%, and 11.4% of the variability in IA, HI, and ODD, respectively), and AL accounted for <1% of the variability in IA, HI, and ODD. The significant variance contributed by TI, DER, and AL suggests that TI is associated with IA, HI, and ODD. As DER and AF also contributed to the prediction of IA, HI, and ODD above and beyond TI, this means that difficulties in emotional regulation are also associated with IA, HI, and ODD. However, the relatively high variance contributed by TI and the relatively low variance contributed by DER and AL after controlling for TI can be interpreted as indicating that TI has more associations than difficulties in emotional regulation with IA, HI, and ODD.

Our findings supporting the associations for TI and emotional regulation difficulties with IA, HI, and ODD are consistent with the existing findings and the theory that suggests that ADHD [2,5,53,54,55] and ODD [4,10,12,13,19,27] are disorders of impulsivity or poor response inhibition. Indeed, according to the TIH, preschool-aged children with temperaments reflecting impulsivity (high irritability, negative affectivity, and poor inhibitory control) first show the development of HI symptoms, and then progress to ODD when appropriate risk factors are present. Interestingly, difficulties in emotional regulation have been proposed as one risk factor [2]. Considering this, we have chosen to interpret out findings as suggesting that although difficulties in emotional regulation are a significant predictor of ADHD and ODD, it is better viewed as an important correlate [11,12,13] rather than a core symptom of ADHD and ODD, as suggested by some [10,11,12]. Expressed differently, difficulties in emotional regulation are unlikely to be a diagnostic indicator for ADHD.

Considering DER and AL, our findings showed that when DER was entered before AL (incremental validity analysis 1), AL contributed 1%, 4%, and 0% to the prediction of IA, HI, and ODD above and beyond age, TI, and DER. DER contributed 7.7%, 9.7%, and 11.4% of the variability in IA, HI, and ODD, respectively, beyond age and TI. When AL was entered before DER (incremental validity analysis 2), DER contributed 3.5%, 7.8%, and 6.2% to the prediction of IA, HI, and ODD, above and beyond age, TI, and AL; and AL contributed 4.3%, 2.1%, and 5.3% of the variability in IA, HI, and ODD, respectively, beyond age and TI. Thus, AL contributed very little additional variance to the predictions of IA, HI, and ODD above and beyond age, TI, and DER, but DER contributed additional variance to the predictions of IA, HI, and ODD above and beyond age, TI, and AL. The findings can be interpreted to mean that DER may have more associations with IA, HI, and ODD than AF.

### 4.1. Practical Implications for Assessment and Treatment

Our findings that difficulties in emotional regulation are not core to ADHD and ODD indicate that this factor should not be used directly for the diagnosis of these disorders. This fact notwithstanding, as the findings showed incremental validity in the prediction of ADHD and ODD, after accounting for TI, it can be argued that it would be useful to assess for difficulties in emotional regulation for individuals referred for ADHD and ODD. In this context, as both DER and AL showed significant correlations with IA, HI, and ODD, this could mean that either the DERS-36 and/or ALS-18 could be used for assessing difficulties in emotional regulation.

In relation to treatment, at a more general level, it is worth noting that treatment focused on emotion dysregulation in ADHD is highly desirable because emotion dysregulation is now regarded as a transdiagnostic symptom and risk factor for several psychiatric disorders [56]. It has also been associated with somatic problems, such as a chronic pro-inflammatory status [57,58]. In relation to treatment approaches, the theoretical model [39] underpinning the DERS-36 is rooted in models of cognitive behavioral therapy (CBT) that propose a central role for experiential avoidance in the onset and maintenance of emotional disturbance (“third-wave” models of CBT). Experiential avoidance refers to intolerance of and maladaptive efforts to avoid (usually negative) emotional experiences, and poor emotion regulation is viewed in terms of difficulties experienced when the individual is unable to behave in a way that does not facilitate the achievement of a priori goals, particularly when faced with negative affect or other strong emotional experiences [59]. This model has been strongly accepted and adopted in research and treatment contexts involving emotional problems [60]. Considering this, as the DERS-36 identifies areas for growth in how individuals respond to their emotions, the incremental validity offered by the DER in the prediction of IA, HI, and ODD offers insights for emotion-related, theory-based treatment and management of ADHD and ODD. Given the content of the DERS-36, it can be speculated that individuals with these disorders could benefit from targeting areas such as nonacceptance of negative emotional responses and distress; problems dealing with difficulties in concentration and accomplishing tasks when upset; difficulties controlling behavior when distressed and feeling overwhelmed; difficulties paying attention and acknowledging emotions; poor strategies for regulating the negative emotions being experienced; and poor ability to understand emotions. In addition, the therapeutic options concerning DER are not limited to CBT (as suggested in line 483), but include mainly DBT, as shown by a recent systematic umbrella review [61].

Although our findings are generally consistent with the TIH, there are a couple of findings that are not consistent with the TIH. First, as the TIH postulates that IA symptoms develop secondary to the HI symptoms [16], it can be expected that TI would show a noticeably stronger association with HI than IA. However, our findings did not show this. Rather, as shown in Table 2, column 2, in absolute terms, TI had a higher association with IA (*R*^2^ = 0.445) than HI (*R*^2^ = 0.300). Considering this, it could be speculated that there is a possibility that IA emerges directly from TI and is not secondary to HI. Second, THI postulates that the emergence of ODD from HI is facilitated by difficulties in emotional regulation [2,17]. Considering this, it can be expected that difficulties in emotional regulation would show a reasonable amount of association with ODD. However, as our finding showed only low to modest associations of DER and AL with ODD (*R*^2^ = 0.114 and 0.000, respectively), it can be argued that the view that difficulties in emotional regulation contribute to the emergence of ODD from HI needs to be reexamined.

Our initial evaluation indicated at least an acceptable fit for a three-factor model, with latent factors for TI, DER, and AL, and there was support for the discriminant validity of the factors in this model. These were conducted as incremental validity is only theoretically meaningful when this property is supported [33]. Additional analysis indicated that IT, DER, AF, IA, HI, and ODD scores correlated significantly and positively with each other. As far as we are aware, the significant correlations for the difficulties of emotional regulation constructs with ODD have not been tested previously. Furthermore, although our incremental validity valuation showed that AL had no associations with IA, HI, or ODD above and beyond TI and DER, it was associated with them when considered alone. The correlation findings also indicated that AL was associated negatively with age and showed no associations with gender. These findings for age and gender are consistent with past findings that have shown that that scores on the DERS-36 decrease with age [47,48,49] and gender has no association with DERS-36 scores [39,50,51,52].

### 4.2. Summary and Conclusions

In summary, the findings in the present study showed that that both TI and difficulties are relevant to ADHD and ODD. However, TI offered a greater contribution in terms of explaining the symptoms of ADHD and ODD, thereby suggesting that, while TI is core to ADHD, difficulties in emotional regulation are important, but unlikely to be core to ADHD and ODD, i.e., difficulties in emotional regulation are unlikely to represent a diagnostic indicator for ADHD. While these findings may not be considered new, they do add some clarity to the controversies surrounding this area, and they are novel in that way. To some degree, we were able to add clarity because we used the model-based SEM approach proposed by [33] to examine incremental validity, which is clearly a strength of the current study. Unlike the more traditional hierarchical regression approach to testing incremental validity, the SEM approach is considered better because it uses latent scores that are free of random measurement errors that could bias the findings [33]. Also, we were able to test for incremental simultaneously with multiple predictors and outcomes in the same model.

#### Limitations

Although the current study delivers valuable information on the associations of TI and difficulties in emotional regulation with ADHD and ODD in adults, the findings and interpretations need to be considered with several limitations in mind. First, emotional regulation ratings are potentially influenced by factors such as age, gender, culture, and existing psychopathologies [62,63,64,65]. Not controlling for culture and existing psychopathologies in the present study may have confounded the findings. Second, as all participants in this study were from the general community and not selected randomly, our findings may be further confounded and limited in terms of generalization, including their application to those with the potential for clinical levels of different psychopathologies. Third, as all data used were collected using self-rating questionnaires, it is possible that the ratings were confounded by common method variance. Fourth, we cannot be certain as to whether our findings will be replicated with other measures of TI, DER, and AL. Fifth, our findings were obtained from a single study, and therefore, replication is essential. Sixth, while we established sufficient power for the study, it may still be possible that the findings would have been different if the sample had been larger. Seventh, following the tutorial example [33], we did not adjust for multiple comparisons, thereby controlling the possibility of Type 1 error. Eighth, emotion-regulating circuits may mature until 25 years of age. Thus, our failure to examine age subgroups (e.g., <25 vs. >25 years) is another limitation. Given these limitations, further research controlling for the limitations noted above is necessary. Nevertheless, despite the limitations mentioned, the findings of this study can be expected to contribute significantly to theory and clinical practice regarding emotional regulation, ADHD, and ODD in adults. It is recommended that clinicians and researchers consider the findings and interpretations from this present study when integrating information on emotional regulation for the treatment of ADHD and ODD.

## Figures and Tables

**Figure 1 behavsci-14-00598-f001:**
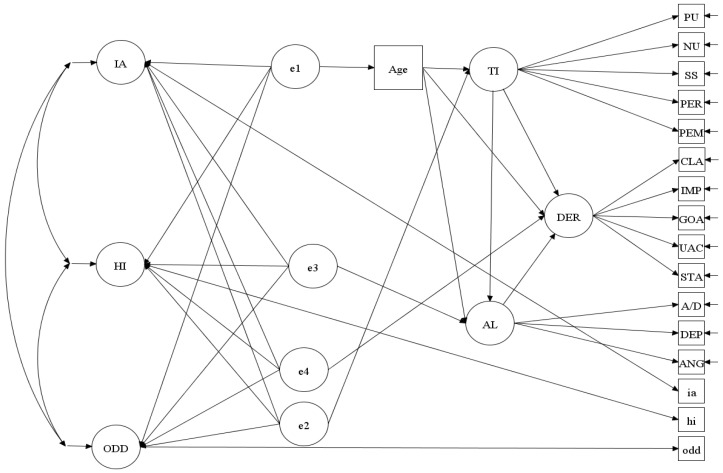
Structural model diagram for the incremental validity (in sequence) of trait impulsivity, dysfunctional emotional regulation, and affect lability predicting IA, HI, and ODD latent outcomes. Note. ANG = anger; DEP = depression; A/D = anxiety/depression; STA = strategies; UAC = unawareness; IMP = impulse; CLA = clarity; PEM = lack of perseverance; PER = lack of premeditation; SS = sensation seeking, NU = negative urgency; PU = positive urgency; AL = affect lability; DER = dysfunctional emotional regulation; AL = affect lability.

**Table 1 behavsci-14-00598-t001:** Fit values for 1-factor and 3-factor models with factor for trait impulsivity, dysfunctional emotional regulation, and affect lability.

Model	MLR*χ*^2^ (*df*)	CFI	TLI	RMSEA (90% CI)	SRMR
1-Factor	638.736 (65)	0.794	0.753	0.130 (0.121–0.139)	0.076
3-Factor	369.675 (62)	0.889	0.861	0.097 (0.088–0.107)	0.062

*Note*. CI= confidence interval; RMSEA = root mean square error of approximation; CFI = comparative fit index; TLI = Tucker–Lewis Index. SRMR = Standardized Root Mean Square Residual; All MLR*χ*^2^ values were significant (*p* < 0.001).

**Table 2 behavsci-14-00598-t002:** Semi-partial correlation and change in R^2^ for the predictions of IA, HI, and ODD by (in sequence) age, trait impulsivity (TI), dysfunctional emotional regulation (DER), and affect lability (AL).

		Age	TI	DER	AL
IA	*β* (*SE*); z	−0.104 (0.045); 2.311	0.667 (0.041); 16.268	0.277 (0.051); 5.431	0.040 (0.049); 0.816
	*R*^2^ (95% CI)	0.011 (−0.01/0.028)	0.445 (0.382/0.508)	0.077 (0.034/0.120)	0.001 (−0.004/0.006
HI	*β* (*SE*); z	−0.014 (0.045); 0.311	0.548 (0.043); 12.744	0.309 (0.051); 6.059	−0.066 (0.050); 1.320
	*R*^2^ (95% CI)	0.000 (−0.000/0.001)	0.300 (0.2370/0.365)	0.095 (0.048/0.142)	0.004 (−0.001/0.015)
ODD	*β* (*SE*); z	−0.217 (0.037); 5.865	0.573 (0.039) 14.692	0.338 (0.047); 7.191	0.020 (0.041); 0.488
	*R*^2^ (95% CI)	0.047 (0.012/0.089)	0.328 (0.263/0.393)	0.114 (0.063/0.165)	0.000 (−0.002/0.002)

**Table 3 behavsci-14-00598-t003:** Semi-partial correlation and change in *R*^2^ for the predictions of IA, HI, and ODD by (in sequence) age, trait impulsivity, affect lability, and dysfunctional emotional regulation.

		Age	TI	AL	DER
IA	*β* (*SE*); z	−0.104 (0.045); 2.311	0.667 (0.041); 16.268	0.208 (0.043); 4.84	0.188 (0.047); 4.00
	*R*^2^ (95% CI)	0.011 (−0.01/0.028)	0.445(0.382/0.508)	0.043 (0.009/0.077)	0.035 (0.004/0.066)
HI	*β* (*SE*); z	−0.014 (0.045); 0.311	0.548 (0.043); 12.744	0.146 (0.055); 2.655	0.280 (0.048); 5.833
	*R*^2^ (95% CI)	0.000 (−0.000/0.001)	0.300 (0.237/0.365)	0.021 (−0.003/0.045)	0.078 (0.034/0.122)
ODD	*β* (*SE*); z	−0.217 (0.037); 5.865	0.573 (0.039) 14.692	0.230 (0.049); 4.694	0.248 (0.042); 5.905
	*R*^2^ (95% CI)	0.047 (0.012/0.089)	0.328 (0.263/0.393)	0.053 (0.016/0.090)	0.062 (0.022/0.102)

## Data Availability

Date will be made available from the corresponding author upon reasonable request.

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
