# Peer review of "Incremental Validity of Trait Impulsivity, Dysfunctional Emotional Regulation, and Affect Lability in the Predictions of Attention Deficit Hyperactivity Disorder and Oppositional Defiant Disorder Symptoms in Adults"

_behavsci, 2024, doi:10.3390/bs14070598_

Round 1

Reviewer 1 Report

Comments and Suggestions for Authors

Thank you so much for your paper. Congratulations! It is a well-written study. I have only some comments which should be taken into account in order to improve the paper.

Lines 28-30: Please provide more explanations about the interplay of these symptoms. Providing examples of symptoms would be also beneficial. Please just clarify.

The paper uses a lot of acronyms, and this makes it hard to read. It would be beneficial to use only ADHD and ODD. Feel free to use or reject this comment.

A note for figure 1 is extremely long, and this seems inappropriate. Please present the description of the model in the text.

Are you sure that this sign "Γ matrix" should be used? 

It would be nice to present Limitations, Practical Implications, and Conclusions with separate subsections in the Discussion.

Author Response

Reviewer 1

We appreciate Reviewer 1’s kind comments.

Lines 28-30: Please provide more explanations about the interplay of these symptoms. Providing examples of symptoms would be also beneficial. Please just clarify.

Response. We have provided a brief overview on the interplay of the symptoms (p. 2, para 3, yellow highlighted).

The paper uses a lot of acronyms, and this makes it hard to read. It would be beneficial to use only ADHD and ODD. Feel free to use or reject this comment.

Response. Given the option to reject, we have done so. These are generally standard acronyms which are necessary and as such we do not believe the number of them is excessive.

A note for figure 1 is extremely long, and this seems inappropriate. Please present the description of the model in the text.

Response. As requested, we have removed the test referred to for Figure 1 and integrated it into the main document (p. 7, para 3 to p. 8, para 2, yellow highlighted).

Are you sure that this sign "Γ matrix" should be used? 

Response. Yes.

It would be nice to present Limitations, Practical Implications, and Conclusions with separate subsections in the Discussion.

Response. We have conformed to this recommendation (see discussion section).

Reviewer 2 Report

Comments and Suggestions for Authors

Kudos to the authors for making the argument for and analysis of incremental validity accessible for readers.  This was a very well-done paper about a quite nuanced topic (the incremental validity of trait impulsivity over affective lability and difficulties in emotional regulation in predicting oppositional defiant-disorder and symptom clusters of ADHD).  

All this paper is missing is a traditional demographics table outlining how many participants met criteria for ADHD, ODD, etc.  A sense of the raw numbers would be helpful in determining the clinical significance of these effects.  Otherwise, a very fine job by this authorship team.

Comments on the Quality of English Language

Very minor typo correction needed.

Author Response

Reviewer 2

Kudos to the authors for making the argument for and analysis of incremental validity accessible for readers.  This was a very well-done paper about a quite nuanced topic (the incremental validity of trait impulsivity over affective lability and difficulties in emotional regulation in predicting oppositional defiant-disorder and symptom clusters of ADHD).  

Response. Thank you. We appreciate your comments.

All this paper is missing is a traditional demographics table outlining how many participants met criteria for ADHD, ODD, etc.  A sense of the raw numbers would be helpful in determining the clinical significance of these effects.  Otherwise, a very fine job by this authorship team.

Response. We have provided these details in Supplementary Table S1, and in the test (p. 34, para 4, blue highlight).

Very minor typo correction needed.

Response. We have paid special attention to such errors, and made the corrections where needed.

Reviewer 3 Report

Comments and Suggestions for Authors

GENERAL COMMENTS

This original article aims to investigate the relevance of both trait impulsivity (TI) and emotion dysregulation for ADHD and ODD. The authors claim that "TI offered a greater contribution in explaining the symptoms of ADHD and ODD," although it is acknowledged that emotion dysregulation remains an important therapeutic target for these disorders.

Concerning the content, the topic holds relevance within the ongoing discussion about emotion dysregulation in these disorders, but the authors' findings suffer from some limitations (e.g., absence of correction for comorbidities, as mentioned in the Discussion). Integrating relevant literature about the rationale for focusing on emotion dysregulation in ADHD patient management, besides diagnosis, is crucial.

A better and more explicit definition of "core" symptom is needed. From the authors' own results and previous literature, it is clear that emotion dysregulation is extremely relevant for ADHD patient management and prognosis. It is imprecise to say that emotion dysregulation is not a "core" symptom only because it does not predict some ADHD symptoms: emotion dysregulation could be a different dimension (but perhaps equally relevant) of the disorder(s), not captured by the other clinical scores included by the authors. In this sense, it may be found to be "core" if different clinical score characteristics of the disorders had been considered. Please revise the manuscript to include and address this nuance, at least as a Limitation.

Finally, important formal and methodological revisions are also needed. Thus, all points below need to be addressed before the article is deemed suitable for publication.

ABSTRACT

Please report p-values in the abstract for the main results.

Please mention the main Limitations in the abstract.

These two sentences regarding the role of DER seem contradictory for someone that has not read the full article, please revise:

“DER predicted ADHD and ODD symptoms over age and TI”

“TI is core to ADHD, and although DER is important, it is unlikely to be core to ADHD and ODD”

INTRODUCTION

Better explanations of the rationale for choosing the clinical scores selected by the authors are needed.

The manuscript reads: “the core symptoms of Attention Deficit/Hyperactivity Disorder (ADHD) are inattention (IA; 9 symptoms) and hyperactivity/impulsivity (HI; 9 symptoms). For Oppositional Defiant Disorder (ODD) there are 8 symptoms.” However, ODD symptoms besides TI are somehow neglected in the manuscript, which mainly focuses on IA, TI, and HI. Please explain and address this point.

METHODS

-       Please provide in the supplementary material all the questionnaires used (with translations if necessary).

-       Please provide further details on multiple comparisons adjustment.

RESULTS

Results are overall well presented, thank you.

DISCUSSION

The Discussion overall provides interesting information, but a critical and more in-depth discussion of why it is crucial to focus specifically on emotion dysregulation in ADHD is needed, e.g. expanding the paragraph starting at line 482, on the clinical relevance of improving emotion dysregulation, or in the Introduction. For instance, the authors may briefly mention that emotion dysregulation is a transdiagnostic symptom and risk factor for several psychiatric disorders, and which has also been associated to somatic problems, such as a chronic pro-inflammatory status (e.g. see recent reviews on the topic: “Inflammation and emotion regulation: a narrative review of evidence and mechanisms in emotion dysregulation disorders”, DOI: 10.1042/NS20220077; “A systematic review of associations between emotion regulation characteristics and inflammation” DOI: 10.1016/j.neubiorev.2023.10516). Importantly, the therapeutic options concerning DER are not limited to CBT (as suggested at line 483), but include mainly DBT, as shown by a recent systematic umbrella review (refer to: “Interventions targeting emotion regulation: A systematic umbrella review”, DOI: https://doi.org/10.1016/j.jpsychires.2024.04.025). Importantly,

Hence, despite the authors’ findings that emotion dysregulation may not be an ADHD “core” symptom, these points and the authors’ own results point to the crucial importance of emotion dysregulation in the management of ADHD patients.

LIMITATIONS

Limitations should be expanded based on the points above, where relevant, especially concerning the Methods.

Emotion-regulating circuits may mature up until 25 years of age. Please include the fact that no sub-analysis on age subgroups (e.g. <25 vs >25 years) was done as a limitation, if not possible.

MINOR COMMENTS

Please correct typos throughout the manuscript

“Author Contributions” are incomplete

Please add an explanation of the abbreviations throughout the manuscript upon first use, and also in the captions: tables should be interpretable independently from the text.

Comments on the Quality of English Language

Please thoroughly revise the English, as there are English errors that make comprehension harder, e.g.:

-       I do not think the comma is appropriate in this sentence “The ADHD and ODD literature, depict a separation of difficulties in emotion regulation” (63)

-       Additionally, I would suggest using “depicts” since literature is uncountable.

-       These are just examples, many other points remain

Author Response

Reviewer 3

GENERAL COMMENTS

This original article aims to investigate the relevance of both trait impulsivity (TI) and emotion dysregulation for ADHD and ODD. The authors claim that "TI offered a greater contribution in explaining the symptoms of ADHD and ODD," although it is acknowledged that emotion dysregulation remains an important therapeutic target for these disorders.

Concerning the content, the topic holds relevance within the ongoing discussion about emotion dysregulation in these disorders, but the authors' findings suffer from some limitations (e.g., absence of correction for comorbidities, as mentioned in the Discussion). Integrating relevant literature about the rationale for focusing on emotion dysregulation in ADHD patient management, besides diagnosis, is crucial.

Response. We have provided this information (focusing on emotion dysregulation in ADHD patient management) in the test (p. 11, para 5, green highlighted).

A better and more explicit definition of "core" symptom is needed. From the authors' own results and previous literature, it is clear that emotion dysregulation is extremely relevant for ADHD patient management and prognosis. It is imprecise to say that emotion dysregulation is not a "core" symptom only because it does not predict some ADHD symptoms: emotion dysregulation could be a different dimension (but perhaps equally relevant) of the disorder(s), not captured by the other clinical scores included by the authors. In this sense, it may be found to be "core" if different clinical score characteristics of the disorders had been considered. Please revise the manuscript to include and address this nuance, at least as a Limitation.

Response. We have described what we mean by “core” throughout the test, starting on p. 4, para 1 (green highlighted).

Finally, important formal and methodological revisions are also needed. Thus, all points below need to be addressed before the article is deemed suitable for publication. 

ABSTRACT

Please report p-values in the abstract for the main results.

Response. Complying with this request would mean presenting 18 p values. We believe that this could be highly distractive and could confuse readers. We have therefore not reported p values in the abstract.

Please mention the main Limitations in the abstract.

Response. We have done so (green highlighted in the abstract).

These two sentences regarding the role of DER seem contradictory for someone that has not read the full article, please revise:

“DER predicted ADHD and ODD symptoms over age and TI”

“TI is core to ADHD, and although DER is important, it is unlikely to be core to ADHD and ODD”

Response. We have revised these sentences to increase clarity.

INTRODUCTION

Better explanations of the rationale for choosing the clinical scores selected by the authors are needed.

Response. We are somewhat confused why this was recommended (and how to respond to it) as almost the entire introduction was devoted to this very issue.

The manuscript reads: “the core symptoms of Attention Deficit/Hyperactivity Disorder (ADHD) are inattention (IA; 9 symptoms) and hyperactivity/impulsivity (HI; 9 symptoms). For Oppositional Defiant Disorder (ODD) there are 8 symptoms.” However, ODD symptoms besides TI are somehow neglected in the manuscript, which mainly focuses on IA, TI, and HI. Please explain and address this point.

Response. Again, we are confused why this was recommended because the very first paragraph and almost the introduction covered this issue

METHODS

Please provide in the supplementary material all the questionnaires used (with translations if necessary).

Response. As requested, all but the Current Symptom Checklist are included in the supplementary. As mentioned in the test. the Current Symptom checklist was not provided as it is protected by copyright (p. 5, para 3, green highlight).

Please provide further details on multiple comparisons adjustment.

Response.  As mentioned in the text, although there were multiple comparisons involved, we did not adjust for this, following the tutorial example in Fan and Hancock (2022). (See p.8, para 2, green highlighted.)

RESULTS

Results are overall well presented, thank you.

Response. Thank you.

DISCUSSION

The Discussion overall provides interesting information, but a critical and more in-depth discussion of why it is crucial to focus specifically on emotion dysregulation in ADHD is needed, e.g. expanding the paragraph starting at line 482, on the clinical relevance of improving emotion dysregulation, or in the Introduction. For instance, the authors may briefly mention that emotion dysregulation is a transdiagnostic symptom and risk factor for several psychiatric disorders, and which has also been associated to somatic problems, such as a chronic pro-inflammatory status (e.g. see recent reviews on the topic: “Inflammation and emotion regulation: a narrative review of evidence and mechanisms in emotion dysregulation disorders”, DOI: 10.1042/NS20220077; “A systematic review of associations between emotion regulation characteristics and inflammation” DOI: 10.1016/j.neubiorev.2023.10516). Importantly, the therapeutic options concerning DER are not limited to CBT (as suggested at line 483), but include mainly DBT, as shown by a recent systematic umbrella review (refer to: “Interventions targeting emotion regulation: A systematic umbrella review”, DOI: https://doi.org/10.1016/j.jpsychires.2024.04.025). Importantly,

Response. The transdiagnostic implications for treatment are now covered (p. 11, para 5, green highlighted). The therapeutic options concerning DER is also covered (p.11, para 1, green highlighted).

Hence, despite the authors’ findings that emotion dysregulation may not be an ADHD “core” symptom, these points and the authors’ own results point to the crucial importance of emotion dysregulation in the management of ADHD patients.

LIMITATIONS

Limitations should be expanded based on the points above, where relevant, especially concerning the Methods.

Response. Included where necessary throughout.

Emotion-regulating circuits may mature up until 25 years of age. Please include the fact that no sub-analysis on age subgroups (e.g. <25 vs >25 years) was done as a limitation, if not possible.

Response. Included (p. 13, para 2, green highlighted).

MINOR COMMENTS

Please correct typos throughout the manuscript

“Author Contributions” are incomplete

Response. Completed now (green highlighted).

Please add an explanation of the abbreviations throughout the manuscript upon first use, and also in the captions: tables should be interpretable independently from the text.

Response. Done so throughout.

Comments on the Quality of English Language

Please thoroughly revise the English, as there are English errors that make comprehension harder, e.g.:

-       I do not think the comma is appropriate in this sentence “The ADHD and ODD literature, depict a separation of difficulties in emotion regulation” (63)

-       Additionally, I would suggest using “depicts” since literature is uncountable.

-       These are just examples, many other points remain

Response. The English has been examined careful, with input from two native speakers of English who are professors of psychology, holding PhDs.

As you will notice we have addressed all the comments and suggestions made by you and the reviewers. We hope that the changes meet with your satisfaction, and we look forward to hearing from you soon.

Thank you.

Yours sincerely,

Authors

Round 2

Reviewer 3 Report

Comments and Suggestions for Authors

They address appropriately all my concerns

Comments on the Quality of English Language

OK